# Health literacy and its determinants among higher education students in the Alentejo region of southern Portugal—A cross-sectional survey

**Jorge Rosário**[1,2,3]* , **Sara Simões Dias**[3,4,5] , **Sónia Dias**[6] , **Ana Rita Pedro**[6]

**1** Polytechnic Institute of Beja, Beja, Portugal, **2** Institute for Research and Advanced Training, University of Évora, Évora, Portugal, **3** Comprehensive Health Research Centre, CHRC, University of Évora, Évora, Portugal, **4** ciTechCare - Center for Innovative Care and Health Technology, Polytechnic of Leiria, Leiria, Portugal, **5** School of Health Sciences, Polytechnic of Leiria, Leiria, Portugal, **6** NOVA National School of Public Health, Public Health Research Centre, Comprehensive Health Research Center, CHRC, NOVA University Lisbon, Lisbon, Portugal

☯ These authors contributed equally to this work.
* Jorge.olhoazul@ipbeja.pt

## Abstract

### Introduction

The capacity of higher education students to comprehend and act on health information is a pivotal factor in attaining favourable health outcomes and well-being. Assessing the health literacy of these students is essential in order to develop targeted interventions and provide informed health support. The aim of this study was to identify the level of health literacy and to analyse its relationship with determinants such as socio-demographic variables, chronic disease, perceived health status, and perceived availability of money for expenses among higher education students in the Alentejo region of southern Portugal.

### Methodology

An observational, descriptive and cross-sectional study was conducted between 22 June and 12 September 2023. An online structured questionnaire consisting of the Portuguese version of the European Health Literacy Survey Questionnaire—16 items (HLS-EU-PT-Q16), including socio-demographic data, presence of chronic diseases, perceived health status, and availability of money for expenses. Data were analysed using independent samples t-test, one-way ANOVA, post-hoc Gabriel's test, and multivariate logistic regression analyses at a significance level of 0.05. Regression models were used to investigate the relationship between health literacy and various determinants. The study protocol was approved by the Ethics Committee of the University of Évora, and all participants gave written informed consent.

**Data Availability Statement:** All relevant data are within the manuscript.

**Funding:** This research was funded by the Foundation for Science and Technology (FCT,

Portugal) through national funds for the REAL Associated Laboratory in Translation and Innovation for Global Health (LA/P/0117/2020). No additional external funding was received for this study. The funder had no role in study design, data collection and analysis, decision to publish, or preparation of the manuscript.

**Competing interests:** The authors have declared that no competing interests exist.

## Results

Analysis of the HLS-EU-PT-Q16 showed that 82.3% of the 1228 students sampled had limited health literacy. The mean health literacy score was 19.3 ± 12.8 on a scale of 0 to 50, with subscores of 19.4 ± 13.9 for health care, 19.1 ± 13.1 for disease prevention, and 19.0 ± 13.7 for health promotion. Significant associations were found between health literacy and several determinants. Higher health literacy was associated with the absence of chronic diseases. Regression analysis showed that lower health literacy was associated with not attending health-related courses, not living with a health professional, perceiving limited availability of money for expenses, and having an unsatisfactory health status.

## Conclusion

This study improves the understanding of health literacy levels among higher education students in Alentejo, Portugal, and identifies key determinants. Higher education students in this region had relatively low levels of health literacy, which may have a negative impact on their health outcomes. These findings highlight the need for interventions to improve health literacy among higher education students and to address the specific needs of high-risk subgroups in the Alentejo.

## Introduction

Health Literacy (HL), as defined by Sørensen et al. (2012), refers to individuals' knowledge, motivation, and competencies to access, understand, evaluate, and apply health information in order to make informed judgements and decisions in everyday life concerning healthcare, disease prevention, and health promotion aimed at maintaining or improving quality of life throughout their lives [1]. It is associated with the cognitive and social skills, essentials to the motivations and ability of individuals to access, understand and use information in ways that promote and maintain good health [2]. Recognising its pivotal role, both the United Nations and the World Health Organization (WHO) highlight the importance of health literacy in achieving the Sustainable Development Goals (SDG3) on health and promoting community well-being [3, 4]. The European Health Literacy Project (HLS-EU) emphasises health literacy as a determinant of health, integrating public health perspectives and individual approaches [1].

These global perspectives are based on the dimensions of health literacy described by Nutbeam (2000): functional, interactive and critical [5]. The functional dimension pertains to the capacity to read and comprehend health-related information and sometimes includes numeracy skills, such as the application of mathematics in daily life [5]. The interactive literacy integrates advanced literacy and cognitive skills with social skills, enabling individuals to actively engage with health information and adapt to changing circumstances [5]. Critical literacy comprises the most advanced skills, allowing individuals to critically analyse information and use it to exert greater control over significant events and situations [5].

In order to gain further insight into the pathways linking health literacy to health behaviours and outcomes, Sørensen et al. (2012) developed the Integrated Model of Health Literacy. This theoretical framework delineates the pathways linking health literacy to health behaviours and outcomes. The model identifies the determinants and factors influencing health literacy levels (the antecedents) and their subsequent health outcomes (the consequences) [1]. This theoretical framework, predominant in Europe, describes four core competencies of health

literacy: access (the ability to seek, find, and obtain health information), understand (the ability to comprehend accessed health information), appraise (the ability to interpret, filter, judge and evaluate the health information that was assessed) and apply (the ability to communicate and utilise the information to make decision to maintain and improve health). The four competencies, namely accessing, understanding, appraising and applying, can be applied across the three domains of the health continuum, including healthcare, disease prevention and health promotion [1].

An understanding of these competencies, serve to highlight the dynamic and holistic nature of health literacy. This encompasses not only the abilities of individuals, but also the inherent complexity of health systems. This framework facilitates comprehension and improvement of health literacy across heterogeneous populations and healthcare settings. The empowerment of individuals through the acquisition of these skills fosters autonomy and informed decision-making in matters pertaining to health [6] thereby enhancing social participation and well-being. Furthermore, the structural organisation and availability of resources have a significant impact on health literacy levels and individuals' ability to make informed health decisions, which in turn affect personal and community health [7].

Recent European surveys on health literacy show that levels of health literacy vary across the population and highlight the links between higher health literacy and better health outcomes, reduced health inequalities and reduced economic burden [8–11]. Health literacy outperforms many socioeconomic factors as a predictor of health status [12, 13], and is recognised as a key determinant within health systems [14, 15]. Low levels of health literacy are associated with an increased risk of mortality [16, 17], more frequent use of hospital emergency services, primary care centres, and general practitioner consultations [18–20], lower use of preventive services [21], and lower quality of life [22].

In the context of Portugal, studies have shown significant differences in health literacy among higher education students. Amaral et al. (2021) reported that 61% of students in Viseu, an interior region of Portugal, exhibited problematic or inadequate health literacy [23], while Pedro et al. (2022) found that 44% of higher education students nationally faced similar challenges [24]. Higher health literacy tends to be associated with higher family income and parental education levels.

Given their pivotal role, higher education students are crucial in shaping health knowledge and practice [25–27]. Therefore, it is crucial to implement targeted training initiatives to improve health literacy among higher education students [28]. Their knowledge, attitudes, and behaviours not only impact their personal lives but also have societal implications. The level of knowledge among higher education students regarding health issues can significantly influence the efficacy of health education [25, 26]. Understanding the health literacy among higher education students in certain regions, such as Alentejo, is critical due to educational and health inequalities.

Transitioning from a national perspective to a more focused regional context, the Alentejo faces particular challenges. It has the lowest percentage of tertiary educated and high drop-out rates [29]. The Alentejo region in southern of Portugal, which is classified both as a Territorial Level 2 (TL2) and a Nomenclature of Territorial Units for Statistics II (NUTS II) region, covers an area of 31,605 km$^2$, about 30% of the national territory. This region has a low population density and is threatened by depopulation due to a decreasing young population, an increasing elderly population and low immigration [30, 31]. Addressing these inequalities requires promoting health literacy and healthy behaviours from an early age, particularly in rural areas facing demographic challenges [32, 33].

The regional focus not only demonstrates the existence of educational and health inequalities but also emphasis the pressing necessity to conduct research and implement solutions at the

local level. By elucidating the health literacy levels of higher education students in Alentejo, this research can provide insights into regional inequalities and inform policies aimed at reducing these disparities, ultimately benefiting both individual and community health outcomes.

It is therefore imperative to assess the level of health literacy among higher education students and investigate its association with various determinants. To the best of our knowledge, the current level of health literacy among higher education students in Alentejo is unknown, which makes it challenging to design effective health interventions.

This study will address the following research questions: What is the level of health literacy among higher education students in the Alentejo region of southern Portugal? What is the relationship between health literacy and determinants such as socio-demographic variables, the presence of chronic disease, perceived health status, and perceived availability of money for expenses among higher education students in the Alentejo region? The responses to these questions will facilitate the implementation of targeted interventions and policies designed to reduce inequalities and enhance health outcomes in this region. Health students, older students, those living with health professionals and those with enough money for expenses are expected to have higher health literacy. The objective of this study is to identify the level of health literacy and to analyse its relationship with determinants such as socio-demographic variables, the presence of chronic disease, perceived health status, and perceived availability of money for expenses among higher education students in the Alentejo region of southern Portugal. This study aims to respond to the significant gap in knowledge about health literacy among higher education students in the Alentejo region.

## Materials and methods

### Study design and setting

An online survey was conducted between 22 June and 12 September 2023. Four public higher education institutions (academic institutions) in Alentejo (a region in the south of Portugal) were selected, corresponding to the statistical territorial unit (NUT II), namely the Polytechnic Institute of Beja, Polytechnic Institute of Portalegre, Polytechnic Institute of Santarém, and University of Évora. Following the necessary authorizations from the University of Évora's rectorate and the presidencies of each institute, an email invitation to participate in the study (including the questionnaire link) was disseminated by them to all undergraduate and integrated master's degree students.

### Study population and sample size

The study population comprised all students enrolled in undergraduate and integrated master's degree programs. Students from all academic years were considered. Postgraduate students were excluded. A total of 13135 students were enrolled for the 2022–2023 academic year [34].

A minimum sample size of 384 was determined based on sample size calculation using a 95% confidence level, a population proportion of 50%, an estimated population size of 13135 (enrolments in the academic year 2022/2023), and a margin of error of 5%. To reduce the margin of error to 3%, a sample of 952 elements was determined, increased by 20% to account for potential non-responses (1143). The sample comprised 1228 students, exceeding the required size to achieve a 3% margin of error. Based on the calculations outlined, the sample of 1228 students was considered representative. This sample size was initially set to achieve a 3% margin of error, thus exceeding the minimum requirements set for statistical confidence and ensuring the robustness of the study's findings regarding higher education students in the Alentejo region.

## Study participants and data collection

The Portuguese version of the 16-item European Health Literacy Survey Questionnaire (HLS-EU-Q16), which used to assess health literacy levels and a basic demographic survey were distributed through a hyperlink using an online survey platform. The survey hyperlink was included in the study invitation. The questionnaire was designed to be completed only once per participant, with no option for multiple submissions to ensure data integrity.

**Institutional approval and promotion.** The governing body of each institution was contacted via email to explain the aims and importance of the study, and to request their approval. Once approval was granted, the institution promoted the link to the questionnaire, along with the consent form, to students via their official social media platforms and email lists. To reinforce and encourage participation, course coordinators were asked to highlight the importance of the survey in emails and meetings. This multi-channel approach was designed to ensure high visibility and maximise response rates.

**Survey administration.** Responses were collected electronically using an online survey tool that automatically recorded and securely stored data. The tool ensured anonymity and confidentiality in line with ethical research standards. Regular reminders were sent to students to maximise participation, and technical support was provided to resolve any problems encountered when completing the survey. The survey was designed to be mobile-friendly, ensuring accessibility on various devices, including smartphone or tablets. It was conciseness to minimise the time required for completions.

**Addressing potential biases.** The potential biases inherent in the online data collection method were recognised, including the exclusion of students without internet access, electronic devices or familiarity with digital technologies. To mitigate these concerns, the survey was made accessible on various devices available at higher education institutions, and participants were provided with technical support available at institutions where necessary. In addition, the multi-channel promotion strategy aimed to widen participation across different student populations, thus ensuring a more diverse and representative sample.

## Measurements

The first section of the questionnaire comprised the presentation of the study's participation conditions and the request for informed, voluntary, and clear consent along with confirmation of age over 16 years. Only after affirmative responses to these two requests, it would be possible to proceed to answer the remaining sections of the questionnaire.

The second section consisted of the HLS-EU-PT-Q16 scale and another set of questions relating to sociodemographic characteristics. It consisted of 31 questions, with an estimated response time of 10 minutes.

## Health Literacy—HL (dependent variable)

The Portuguese version of the European Health Literacy Survey Questionnaire with 16 items (HLS-EU-Q16) was utilized to assess health literacy. This measure is related to the difficulty/ease of access, comprehension, appraisal, and application of information in the domains of healthcare (7 items), disease prevention (5 items), and health promotion (4 items) [35–37].

Each item on the scale has four possible responses (very easy, easy, difficult, and very difficult) and also the option "don't know/refusal". Each response corresponds to a numerical value: 1—very difficult, 2—difficult, 3—easy, 4—very easy, and 5—don't know/refusal. The score of each item of the scale was calculated only if at least 80% of the items contained valid responses (between options 1 and 4). The option 5 (don't know), it was considered missing.

The mean score was calculated for all items on the scale, subsequently converted into a health literacy index score, respecting the guidelines of the European Health Literacy Consortium.

The calculation of the health literacy index score was performed according to the following formula: (mean-1)*(50/3), where mean corresponds to the average of items on the scale, 1—the minimal value of the mean, 3—the range of the mean, and 50—the chosen maximum value of the new index scores [38–41]. Health literacy levels were created based on the cut-offs: excellent (>42–50), sufficient (>33–42), problematic (>25–33), and inadequate (0–25) [39–41]. Subsequently, the levels were dichotomized into limited (combining inadequate and problematic health literacy levels) and adequate health literacy (combining sufficient and excellent health literacy levels). A pretest was conducted with 32 representative individuals from the target population, with no need for correction of questions or formatting.

The Cronbach's alpha coefficient for HLS-EU-PT-Q16, which indicates internal consistency, was 0.89 overall, and 0.783 for the health care subdomain, 0.724 for the disease prevention subdomain, and 0.703 for the health promotion subdomain [35].

## Independent variables

Sociodemographic variables, academic background, chronic disease, self-perceived health status, and perceived availability of money for expenses were considered.

Sociodemographic variables such as age, gender, country where secondary education was completed, district (inside or outside Alentejo) of secondary education conclusion, displacement from usual place of residence to attend the course, cohabitation with a healthcare professional (father, mother, or person with whom they reside being a healthcare professional), highest educational level of parents or the person with whom the student lived, according to the International Standard Classification of Education—ISCED [42], were included. For analysis, we grouped the education variables as following: no formal education and primary, lower, secondary and university.

Academic background included academic institution, academic year, being a finalist, completion a previous course in the health field, and frequenting a health-related course.

The presence of chronic diseases and self-perceived health status were included [43, 44]. Participants were asked to rate their current health status on a 5-point scale, which was initially classified into six categories: very bad, bad, fair, good, very good, and prefer not to answer. It was then reclassified into satisfactory (good and very good) and unsatisfactory (very bad, bad and fair) health status.

The perceived availability of money for expenses was categorized as rarely, sometimes, almost always, always, or prefer not to answer [37]. Subsequently, it was recategorized into bad (rarely and sometimes) and good (almost always and always).

## Data analysis

The statistical analysis was performed using the IBM-SPSS version 29. Descriptive analysis was conducted: mean and standard deviation (SD) to quantitative variables and frequencies (absolute and percentages) to qualitative variables. A t-test was used for comparing health literacy differences between two groups, since normal distribution and homogeneity of the variances was presented. One-way ANOVA was used for comparing the health literacy differences among multiples groups since there was normal distribution and homogeneity of the variances. After finding a significant result in the overall ANOVA test, Gabriel's test was used to determine which specific groups differ significantly from each other. A multiple linear regression analyses was computed to identify the determinants of health literacy. A statistical significance level of $p < .05$ was considered.

## Ethical considerations

The study was approved by the Ethics Committee of the University of Évora (Document number 22091) and approved by the Scientific Council. All students were informed of the study objectives, which were outlined in a written statement of voluntary and informed consent in the first section of the self-administered questionnaire. Written digital informed consent was obtained from study participants prior to completing the survey form by filling in the appropriate space on the first page. Students who did not consent to participate did not have access to the questionnaire content. Confidentiality, anonymity, and data anonymisation are ensured.

## Results

### Socio-demographic characteristics

The sample comprised 1228 students with an average age of 21.34 ± 2.9 years. Of these students, 54.2% identified as female, 36.5% as male and 9.3% did not specify their gender. Most of the students had completed secondary education in Portugal (97.0%), in an Alentejo district (65.6%), had been displaced from their place of residence (76.1%), had not cohabited with a health professional (94.4%), and had parents who had completed secondary education (39.4%). Only 4.7% of students had completed a health course prior to this study.

In terms of the higher education institution attended, 36.6% of students were affiliated to the Polytechnic Institute of Beja, 34.9% to the University of Évora, 17.5% to the Polytechnic Institute of Santarém, and 11.0% to the Polytechnic Institute of Portalegre. A significant proportion of students were in their third year (31.0%), followed by those in their second (25.0%), fourth (24.7%) and first (15.1%), with the smallest proportion in their fifth year (4.2%). The largest group of students were not finalists (68.0%), had not completed a health-related course before (95.3%), and had not attended a health-related course (66.8%).

Regarding their health status, 68.5% of the students had no chronic diseases. In addition, 76.3% of students reported that their perceived health status was unsatisfactory. The perceived availability of money for expenses (food, housing, education and health) was rated as bad by 77% of the students. Table 1 summarises the differences between the characteristics of higher education students and the mean of the health literacy index.

### Health literacy

Table 2 presents the distribution of observed frequencies, percentages, means and standard deviations of responses to each item on the HLS-EU-PT-Q16 scale. When analysing the absolute and relative frequencies in per cent of the individual response items of the scale, predominant occurrences of "very difficult" responses were observed in item 15 "understand information in the media about how to get healthier?" with 37.0%. Responses categorized as "difficult" were predominantly observed in item 5 "judging when you might need to get a second opinion from another doctor with 43.2%. Conversely, the highest frequencies of "easy" responses were found in item 7 "follow the instructions of your doctor or pharmacist" with 33.5%. Among the options classified as "very easy," the most prevalent selection was observed in item 9 "follow instructions from your doctor or pharmacist?" with 19.5%.

In terms of mean scores (ranging from very easy = 4 to very difficult = 1), the highest score (2.3 ± 1.0) was observed in four items regarding: item 4; item 7; item 9 and item 10. The item with the lowest mean score (1.9 ± 0.8) was item 5.

Fig 1 (Fig 1) displays the distribution of health literacy levels by percentage across general health literacy and domains. It was observed that 62.4% of the students had inadequate levels for general health literacy, while 2.4% had excellent levels. Processing information related to

**Table 1. Differences between the characteristics of higher education students and the mean of the health literacy index (N = 1228).**

| Characteristics of the higher education students—independent variables | | N | % | Mean (SD) [0;50] | $p$ |
|---|---|---|---|---|---|
| Age | [16;20] | 502 | 40.9 | 19.8 (13.1) | $< .001^{\Omega}$ |
| | [21;25] | 566 | 46.1 | 18.1 (12.3) | |
| | [26;30] | 160 | 13.0 | 22.5 (13.2) | |
| Gender (n = 1113) | Feminine | 665 | 59.7 | 22.6 (12.1) | $< .001^{\dagger}$ |
| | Masculine | 448 | 40.3 | 17.1 (12.9) | |
| Country of completion of secondary education | Portugal | 1191 | 97.0 | 24.6 (12.6) | $< .01^{\dagger}$ |
| | Outside of Portugal | 37 | 3.0 | 19.2 (12.8) | |
| District in Portugal where secondary education was completed (n = 1191) | Inside Alentejo | 781 | 65.6 | 19.5 (13.0) | $.135^{\dagger}$ |
| | Outside Alentejo | 410 | 34.4 | 18.6 (12.3) | |
| Displaced from usual residence to attend the course | Yes | 934 | 76.1 | 18.1 (12.6) | $< .001^{\dagger}$ |
| | No | 294 | 23.9 | 23.2 (12.8) | |
| Cohabitation with a health professional | Yes | 69 | 5.6 | 28.5 (8.8) | $< .001^{\dagger}$ |
| | No | 1159 | 94.4 | 18.8 (12.8) | |
| Level of education of the student's parents or the person with whom the student lived | No formal education/primary | 31 | 2.5 | 18.6 (16.3) | $.092^{\Omega}$ |
| | Lower | 394 | 32.1 | 19.7 (12.7) | |
| | Secondary | 484 | 39.4 | 20.1 (12.9) | |
| | University | 319 | 26.0 | 17.8 (12.4) | |
| Academic institution | Polytechnic Institute of Beja | 450 | 36.6 | 26.5 (11.1) | $< .001^{\Omega}$ |
| | Polytechnic Institute of Portalegre | 135 | 11.0 | 29.9 (13.1) | |
| | Polytechnic Institute of Santarém | 215 | 17.5 | 16.3 (11.0) | |
| | University of Évora | 428 | 34.9 | 12.6 (11.1) | |
| Academic year | First | 186 | 15.1 | 29.0 (9.1) | $< .001^{\Omega}$ |
| | Second | 307 | 25.0 | 19.1 (12.6) | |
| | Third | 381 | 31.0 | 13.6 (11.0) | |
| | Fourth | 303 | 24.7 | 21.2 (13.5) | |
| | Fifth | 51 | 4.2 | 17.7 (10.9) | |
| Finalist | Yes | 393 | 32.0 | 20.6 (12.8) | $< .01^{\dagger}$ |
| | No | 835 | 68.0 | 18.7 (12.8) | |
| Previous completion of a course in healthcare | Yes | 58 | 4.7 | 22.3 (11.0) | $< .001^{\dagger}$ |
| | No | 1170 | 95.3 | 19.2 (12.9) | |
| Healthcare-related course | Yes | 408 | 33.2 | 30.0 (7.8) | $< .001^{\dagger}$ |
| | No | 820 | 66.8 | 14.0 (11.5) | |
| Chronic disease | Yes | 387 | 31.5 | 15.2 (12.4) | $< .001^{\dagger}$ |
| | No | 841 | 68.5 | 21.2 (12.6) | |
| Perceived health status (n = 1222) | Satisfactory | 290 | 23.7 | 31.2 (7.4) | $< .001^{\dagger}$ |
| | Unsatisfactory | 932 | 76.3 | 15,6 (11.9) | |
| Perceived availability of money for expenses | Good | 283 | 23.0 | 31.6 (7.1) | $< .001^{\dagger}$ |
| | Bad | 945 | 77.0 | 15.7 (11.9) | |

SD—Standard Deviation,

$\Omega$—ANOVA;

$\dagger$—t student

health promotion appears to be more challenging for students, with 66.4% having inadequate levels compared to 63.2% for disease prevention and 60.3% for healthcare.

Following the categorisation of health literacy levels into two distinct groups, it was found that 82.3% of the students exhibited limited health literacy while 17.7% demonstrated adequate

**Table 2. Distribution of observed frequencies, percentages, mean and standard deviation of responses to each item on the HLS-EU-PT-Q16 scale (N = 1228).**

| Area | On a Scale from Very Easy to Very Difficult, How Easy Would You Say It Is To | 1—"Very difficult" (n, %) | 2—"Difficult" (n, %) | 3—"Easy" (n, %) | 4—"Very easy" (n, %) | Mean (SD) [1;4] | 5—"Don't know/ Refusal" (n, %) [missing] |
|---|---|---|---|---|---|---|---|
| HC | 1. find information on treatments of illness that concern you | 432; 35.2 | 297; 24.2 | 364; 29.6 | 122; 9.9 | 2.1 (1.0) | 13; 1.1 |
| HC | 2. find out where to get professional help when you are ill? | 390; 31.8 | 433; 35.3 | 315; 25.7 | 86; 7.0 | 2.0 (0.9) | 4; 0.3 |
| HC | 3. understand what your doctor says to you? | 375; 30.5 | 327; 26.6 | 396; 32.2 | 123; 10.0 | 2.2 (0.9) | 7; 0.6 |
| HC | 4. understand your doctor's or pharmacist's instruction on how to take a prescribed medicine | 374; 30.5 | 287; 23.4 | 352; 28.7 | 212; 17.3 | 2.3 (1.0) | 3; 0.2 |
| HC | 5. judge when you may need to get a second opinion from another doctor? | 372; 30.3 | 530; 43.2 | 271; 22.1 | 37; 3.0 | 1.9 (0.8) | 18; 1.5 |
| HC | 6. use information the doctor gives you to make decisions about your illness? | 331; 27.0 | 469; 38.2 | 342; 27.9 | 60; 4.9 | 2.1 (0.8) | 26; 2.1 |
| HC | 7. follow instructions from your doctor or pharmacist? | 326; 26.2 | 313; 25.2 | 411; 33.5 | 178; 14.6 | 2.3 (1.0) | 0; 0 |
| DP | 8. find information on how to manage mental health problems like stress or depression? | 370; 30.1 | 523; 42.6 | 267; 21.7 | 51; 4.2 | 2.0 (0.8) | 17; 1.4 |
| DP | 9. understand health warnings about behaviour such as smoking, low physical activity and drinking too much? | 368; 30.0 | 310; 25.2 | 304; 24.8 | 240; 19.5 | 2.3 (1.1) | 6; 0.5 |
| DP | 10. understand why you need health screenings? | 351; 28.6 | 341; 27.8 | 313; 25.5 | 215; 17.5 | 2.3 (1.0) | 8; 0.7 |
| DP | 11. udge if the information on health risks in the media is reliable? | 386; 31.4 | 501; 40.8 | 273; 22.2 | 55; 4.5 | 2.0 (0.8) | 13; 1.1 |
| DP | 12. decide how you can protect yourself from illness based on information in the media? | 351; 28.6 | 472; 38.4 | 342; 27.9 | 56; 4.6 | 2.0 (0.8) | 7; 0.6 |
| HP | 13. ind out about activities that are good for your mental well-being? | 348; 28.3 | 414; 33.7 | 344; 28.0 | 115; 9.4 | 2.1 (0.9) | 7; 0.6 |
| HP | 14. understand advice on health from family members or friends? | 419; 34.1 | 342; 29.9 | 386; 31.4 | 75; 6.1 | 2.1 (0.9) | 6; 0.5 |
| HP | 15. understand information in the media on how to get healthier? | 454; 37.0 | 314; 25.6 | 387; 31.5 | 73; 5.9 | 2.0 (0.9) | 0; 0.0 |
| HP | 16. judge which everyday behaviour is related to your health? | 367; 29.9 | 338; 27.5 | 399; 32.5 | 121; 9.9 | 2.2 (0.9) | 3; 0.2 |

HC—Healthcare, DP—Disease Prevention; HP—Health Promotion; SD—Standard Deviation.

health literacy. In terms of the healthcare domains, limited health literacy was observed to have a higher percentage in disease prevention (78.2%), while adequate health literacy was observed to have a higher percentage in health promotion (25.7%).

The mean general health literacy index was 19.3 ± 12.8, with a range from 0 to 50. The domains of health literacy pertaining to healthcare, disease prevention, and health promotion were observed to be 19.6 ± 13.9, 19.2 ± 13.1, and 19.0 ± 13.7, respectively. The scores for the health literacy index ranged from a minimum of 0 to a maximum of 50.

The frequency analysis of the ability to process health-related information shows a higher percentage of students in the lower literacy levels. Finding and appraising information were considered the most complex aspects, while applying and understanding were considered the most accessible aspects. Inadequate levels were reported by 70.4% of the students for finding information and 69.5% for appraising, compared to 61.6% for applying and 58.2% for understanding.

Table 3 presents the distribution of the mean and standard deviation of the health literacy index (on a scale 0 to 50) according to the competencies of health literacy: finding/accessing, understanding, appraising, and applying. The highest value of the health literacy index was observed in the understanding competency at the level of disease prevention (22.4 ± 17.3) and healthcare (21.4 ± 16.9). The lowest scores were observed in the appraising competencies of health care (17.0 ± 14.6), disease prevention (17.1 ± 15.0), and finding in disease prevention (17.3 ± 14.9). With regard to the health promotion dimension, it is not possible to assess the

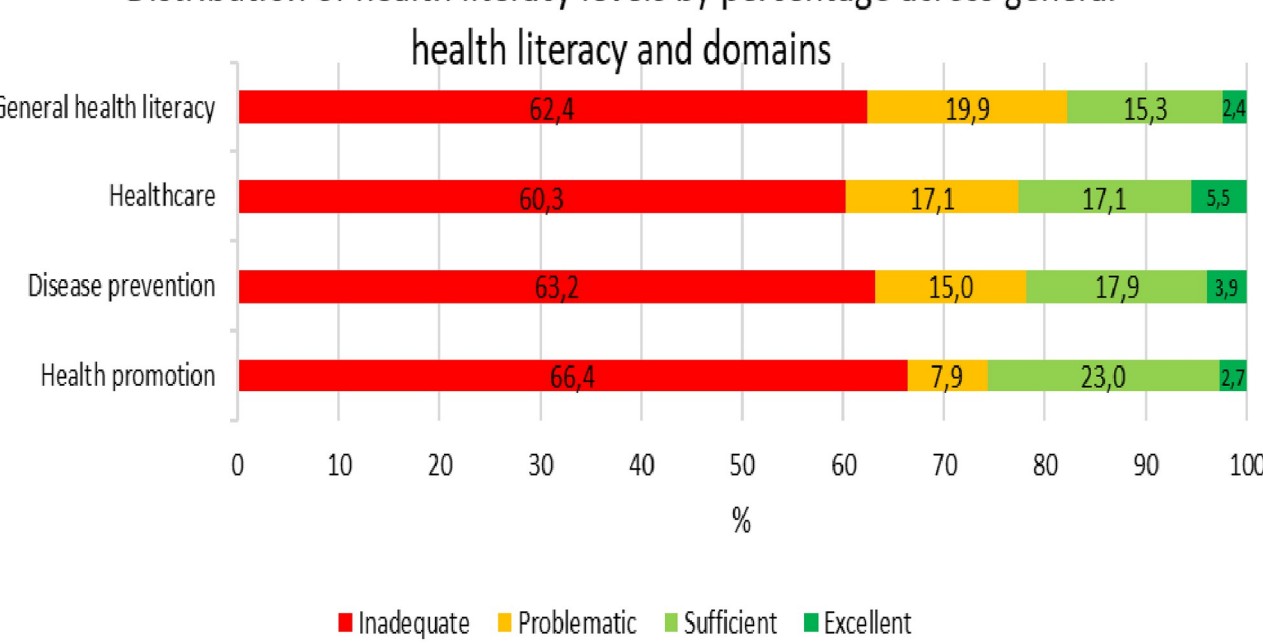

**Fig 1. Distribution of HL levels by percentage across HL and domains.**

competence to apply. In the healthcare dimension, students exhibited greater difficulty in appraising (17.0 ± 14.6). In the domain of disease prevention, it is more challenging to appraise (17.1 ± 15.0) or find information (17.3 ± 14.9) than in the domain of health promotion, where the focus is on understanding (18.1 ± 14.9).

### Differences between independent variables and health literacy

Table 1 highlights significant differences between the characteristics of higher education students and their mean health literacy index. Analysis revealed that students aged 21–25 had a lower mean health literacy index than those aged 26–30. Moreover, students in the 26–30 age group had a higher mean health literacy index compared to those in the 16–20 and 21–25 age groups ($p < .001$). The Gabriel post-hoc test confirmed that students aged 21–25 exhibit a lower mean health literacy index than those aged 26–30. Additionally, students aged 26–30 demonstrate a higher mean health literacy index compared to those aged 16–20 and 21–25 ($p < .01$).

**Table 3. Distribution of the mean and standard deviation of the health literacy index according to the competencies of health literacy: Access, understand, appraisal, and apply (on a scale of 0 to 50).**

| Health literacy dimensions | Health literacy competencies and health literacy index | | | | |
|---|---|---|---|---|---|
| | **Find/Access** | **Understand** | **Appraise** | **Apply** | **All** |
| Healthcare | 18.8 ± 15.4 | 21.4 ± 16.9 | 17.0 ± 14.6 | 21.1 ± 15.3 | 19.6 ± 13.9 |
| Disease Prevention | 17.3 ± 14.9 | 22.4 ± 17.3 | 17.1 ± 15.0 | 18.3 ± 14.7 | 19.2 ± 13.1 |
| Health Promotion | 20.0 ± 16.2 | 18.1 ± 14.9 | 20.5 ± 16.5 | Not available, no data existing | 19.0 ± 13.7 |
| All | 18.2 ± 14.1 | 20.8 ± 16.2 | 17.5 ± 13.6 | 19.5 ± 14.8 | 19.3 ± 12.8 |

The mean health literacy index was significantly higher among female students compared to male students ($p < .001$). Additionally, students who completed secondary education in Portugal had a higher mean health literacy index than those who completed it outside Portugal ($p < .001$). However, no significant differences were observed between the mean health literacy index of students who completed secondary education in Alentejo and those who did not ($p > .05$).

Students who were displaced from their usual residence to attend the course scored lower on the mean health literacy index compared to non-displaced students ($p < .001$). Additionally, students with a father, mother, or cohabitant who was a health professional scored higher on the mean health literacy index than those without such a background ($p < .001$). However, the difference in the mean health literacy index based on the educational level of the students' parents or cohabitants was not statistically significant ($p = .092$). In the context of higher education institutions, students from the Polytechnic Institute of Portalegre exhibited the highest mean health literacy index, followed by those from the Polytechnic Institutes of Beja, Santarém, and the University of Évora ($p < .001$). The Gabriel post-hoc Test indicated that students from the Polytechnic Institute of Beja had a higher mean health literacy index compared to students from the Polytechnic Institutes of Portalegre and Santarém, and the University of Évora ($p < .001$).

Students in the first year of the course had a higher mean of the health literacy index than students in second, third, fourth or fifth academic years ($p < .001$), and the Gabriel post-hoc Test revealed that first year students had higher mean of the health literacy index than the others. Finalists had higher mean of the health literacy index than non-finalists ($p < .001$).

Students with a previous healthcare degree achieved higher mean of the health literacy index than those without ($p < .001$). Additionally, students enrolled in health-related courses had a higher mean of the health literacy index compared to those in non-health-related courses ($p < .001$).

In relation to the perception of health condition, students with chronic conditions had a lower mean health literacy index than those without ($p < .001$). Additionally, students who reported a satisfactory perception of their health status had a higher mean health literacy index compared to those with an unsatisfactory perception ($p < .001$). Concerning perceived availability of money for expenses, students who reported good financial availability had a higher mean health literacy index than those who reported bad availability ($p < .001$).

## The relationship between health literacy and student characteristics using multiple linear regression

The multiple linear regression model presented in Table 4 shows the relationship between health literacy and several independent variables among higher education students in the Alentejo region. The independent variables included were age, presence of chronic disease, attendance in health-related courses, living with a health professional, availability of money for expenses, perceived health status, and academic institution. The intercept represents the baseline health literacy score.

The results show that younger students, particularly those aged 16–20 and 21–25, had significantly lower health literacy scores than those aged 26–30, with differences of 3.5 and 3.4 points respectively ($p < 0.001$). The absence of chronic disease was associated with a higher health literacy score of 1.8 points ($p = 0.002$), suggesting better health literacy among students without chronic disease. Not attending a health-related course was associated with a significantly lower health literacy score of 9.8 points ($p < 0.001$), highlighting the impact of health education on health literacy levels.

**Table 4. Multiple linear regression model.**

| Variables | | β | Standard Error | CI 95% | | Wald Chi-Square | Sig. |
|---|---|---|---|---|---|---|---|
| | | | | Lower | Higher | | |
| Intercept | | 38.5 | 1.67 | 35.2 | 41.7 | 527.9 | < .001 |
| Age | [16;20] | -3.5 | .85 | -5.2 | -1.8 | 17.2 | < .001 |
| | [21;25] | -3.4 | .83 | -5.0 | -1.7 | 16.8 | < .001 |
| | [26;30] | Ref | . | . | . | . | . |
| Chronic disease (no) | | 1.8 | .60 | .6 | 3.0 | 9.1 | .002 |
| Healthrelated course (no) | | -9.8 | .69 | -11.2 | -8.5 | 201.6 | < .001 |
| Cohabitation healthprofissional (no) | | -4.2 | 1.17 | -6.5 | -1.9 | 13.2 | < .001 |
| Availabilty Money for expenses (bad) | | -6.7 | .83 | -8.4 | -5.1 | 66.4 | < .001 |
| Perceived Health status (unsatisfactory) | | -4.3 | .86 | -6.0 | -2.6 | 25.6 | < .001 |
| Academic Institution | Beja | 2.8 | .79 | 1.3 | 4.4 | 12.8 | < .001 |
| | Portalegre | 3.9 | .94 | 2.0 | 5.8 | 17.2 | < .001 |
| | Santarém | 2.0 | .80 | .4 | 3.5 | 6.3 | .012 |
| | Évora | Ref | . | . | . | . | . |

Ref—Reference.

The data also show that not living with a health professional was associated with a lower health literacy score of 4.2 points ($p < 0.001$), suggesting that living with health professionals has a positive impact on health literacy. Poor availability of money for expenses was associated with a lower health literacy score by 6.7 points ($p < 0.001$), highlighting the role of economic factors in health literacy. In addition, perceiving one's health status as unsatisfactory was associated with a lower health literacy score by 4.3 points ($p < 0.001$), indicating a relationship between self-perceived health and health literacy.

Finally, differences in health literacy were observed between academic institutions were observed. Students from Beja had a higher health literacy score by 2.8 points higher than those from Évora ($p < 0.001$), students from Portalegre had a score 3.9 points higher ($p < 0.001$), and students from Santarém had a score 2.0 points higher ($p = 0.012$).

## Discussion

### Summary of findings

This study assessed the level of health literacy among higher education students in Alentejo, located in the south of Portugal, and analysed its relationship with several determinants. The results showed that a significant proportion of students had limited health literacy, which is consistent with previous research on Portuguese populations and highlights the critical need for targeted interventions. This pioneering study of health literacy in the region included 1228 students enrolled in either licentiate or integrated masters programmes, which exceeded the minimum sample size required for the analysis.

### Health literacy levels

The results of the HLS-EU-PT-Q16 questionnaire indicated that 82.3% of students had a limited level of health literacy, while 17.7% had an adequate level. This result is consistent with previous studies [15, 23, 24] that reported a limited level of health literacy among Portuguese higher education students [23, 24]. The level observed in the Alentejo higher education student was almost double that found in studies conducted in Nepal and Ghana [15, 44].

Previous studies conducted in Portugal with the Portuguese population have indicated that there is a limited level of health literacy, with unsatisfactory levels of health literacy observed. A study published by Arriaga et al. identified a general health literacy index of 63.8 ± 11.5 in the general Portuguese population, on a standardised scale of 0 to 100 [37]. Of the participants, 65.0% were classified as having an adequate level of health literacy, while and 22.0% were classified as having a problematic level [37]. A higher proportion of students in higher education in Alentejo exhibited inadequate and problematic health literacy, as evidenced by the observed percentages of 62.4% and 19.9%, respectively, which were higher than those observed in the mainland Portuguese population. A lower level of health literacy is associated with a range of adverse health outcomes, including suboptimal health choices, limited understanding of health conditions, difficulties in reading and comprehending medical information, and the inability to engage in preventive health measures [19, 23]. It can be posited that students enrolled in higher education in the Alentejo region may be more susceptible to adverse health outcomes and suboptimal health decisions due to their limited health literacy.

## Subdomain analysis

The mean health literacy index score for students enrolled in higher education institutions in the Alentejo region was 19.3 ± 12.8 (on a scale of 0 to 50). The mean values for the subdomains were 19.4 ± 13.9 for healthcare, 19.1 ± 13.1 for disease prevention, and 19.0 ± 13.7 for health promotion. The observed values in this study were found to be lower than those reported in previous studies of Portuguese nursing students [23] and students from other disciplines [24].

In terms of processing health-related information, such as finding, understanding, appraising, and applying it, students demonstrated a high health literacy index for understanding information about disease prevention but a lower index for appraising information about healthcare and disease prevention. To the best of our knowledge, no previous studies allow for a comparison of the health literacy index of skills among higher education students in the Alentejo region. However, compared to the general population, Alentejo higher education students exhibited lower scores [37].

## Determinants of health literacy

**Socio-demographic variables.**   Most students were aged between 21 and 25 years, with a mean age of 21.3 ± 2.9 years. Interestingly, the highest level of health literacy was observed among the oldest students, a finding consistent with previous studies assessing health literacy levels among higher education students [23, 24, 27, 45–47]. In terms of gender, 46.1% of students were male, and 54.2% were female. Although there was a statistically significant difference in the mean health literacy index by gender, it did not emerge as an influencing factor on the score. These findings align with previous studies by Kühn et al. and Evans et al., which have demonstrated a relationship between gender and health literacy [25, 44].

A high proportion of students (97.0%) completed their secondary education in Portugal, with 65.6% specifically in the Alentejo region. The proportion of students who completed secondary education outside Portugal is below the expected level, highlighting the challenge of internationalizing higher education institutions. However, this study's findings do not provide evidence to support the impact of internationalization on health literacy.

A discrepancy in health literacy was identified among students who completed secondary education in Portugal, emphasizing the need for further investigation into the health literacy of students who completed secondary education abroad and the factors associated with it. These data indicate the necessity for health literacy interventions for students who completed secondary education outside of Portugal and are currently pursuing higher education in

Portugal. Those with limited health literacy may be more susceptible to making poor health decisions.

No statistically significant differences were observed between students who completed their secondary education in Alentejo and those who completed it outside of Alentejo. It was determined that this phenomenon was not limited to a specific region but rather a national one, and that it was not confined to a particular demographic. The mean health literacy index score for displaced students was found to be lower than that of non-displaced students. It was demonstrated that students who cohabited with health professionals exhibited a higher mean health literacy index score. It was concluded that health literacy interventions should be promoted in the academic context due to the health vulnerability associated with low health literacy. This may be related to the need of students to live temporarily or permanently in another district due to course frequency, without the support of family and friends.

The results of this study indicate that there were no statistically significant differences between the mean health literacy score and the educational level of the students' parents or the person with whom the students lived. This association was contrary to that identified by Zhang et al. (2016) in a study with nursing students [12] or by Cheong et al. (2024) developed with health science students [46, 48] or by Kühn et al. (2022) in a systematic review of the literature that identified it as a determinant of health literacy [25].

The mean health literacy index score for students from the Polytechnic Institute of Portalegre was found to be higher than that of students from other academic institutions. First-year students from all academic institutions exhibited a higher mean health literacy index score. The mean health literacy index score for finalists was found to be higher than that of non-finalists. It is our understanding that it is not possible to make a comparison of these scores due to the lack of studies on health literacy among higher education students in the Alentejo region. However, the outcomes were comparable to those observed in other studies conducted with higher education students [24, 45, 46].

Students enrolled in health-related courses exhibited higher health literacy index values compared to students enrolled in non-related courses. Attending a health-related course is associated with a higher level of health literacy [23, 25, 46]. In a comparative study of higher education Portuguese students in nursing and other areas, Amaral et al. (2021) observed that 61% of students had a problematic or inadequate level of general health literacy [23]. The majority of respondents reported sufficient or problematic levels across all domains of the questionnaire, with nursing students exhibiting a higher level of health literacy and student workers a lower level [23]. This finding was consistent with those observed in other countries [47], including Ghana [44], China [13, 48] and the United States of America [49]. The incorporation of health-related themes in their courses may explain this result, which could positively influence students' health literacy. Therefore, it is crucial to implement health literacy interventions in higher education to enhance students' healthcare, disease prevention and health promotion knowledge, as these factors have a positive impact on their health outcomes [25, 50]. Since the inception of health-related programmes, students have been exposed to health literacy interventions. These interventions seek to familiarise students with health-related knowledge, the healthcare environment, the importance of disease prevention, and health promotion. In this context, it was observed that there were differences in health literacy between students who had previously taken a health-related course and those who had not.

**Chronic disease.** Students with chronic disease exhibited a lower mean health literacy index score than students those without, and students with bad perceived health status also exhibited lower scores than students with good perceived health status. The results were found to be inconsistent with those reported in the study by Pedro et al. (2022), which was carried out with higher education students in Portugal [24]. With regard to the correlation between

perceived health status and the average score on the health literacy index, the results were found to be consistent with those observed by Bánfai-Csonka et al. (2022) in a study of university health science students of different nationalities [51].

**Perceived availability of money for expenses and health status.** The findings of this study indicate that students who perceive limited availability of money for expenses exhibit lower levels of health literacy. It is important to recognise that social and economic inequalities have a significant impact on health literacy, and therefore require particular attention [12, 15, 25, 50–52]. It is important to note that health literacy interventions can be tailored to meet the specific needs of students in relation to the various health literacy subdomains. Health literacy interventions have been demonstrated to impact health outcomes [25, 50].

Regression analysis showed that lower levels of health literacy were associated with several factors, including not attending health-related courses, not living with a health professional, perceiving bad availability of money for expenses, and having an unsatisfactory health status. Other authors have also identified similar outcomes, indicating that students with higher levels of health literacy were enrolled in health-related courses, resided with a healthcare professional, had convenient access to financial resources for expenses, and demonstrated satisfactory health status [25, 45].

**Strengths and limitations.** A critical evaluation of the study's findings reveals several potential biases, limitations and implications for future research. One potential bias is sampling bias, where the sample, although representative with a margin of error of 3%, may not fully represent the diversity of the higher education student population in the Alentejo region. Although efforts were made to reach a wide audience through multiple dissemination channels, inherent biases in voluntary response rates may persist.

In addition, response bias could be an issue as students who chose to participate may have different characteristics to those who did not, potentially skewing the results. Mitigation efforts included regular reminders and incentives to increase participation to reduce non-response bias. Another concern is the digital divide, where differences in access to digital resources may have affected the participation rates and the representativeness of the sample. Although the survey was designed to be mobile-friendly and institutions provided access to the necessary technology, some inequality may still exist.

The focus of the study on the Alentejo region limits the generalisability of the findings to other regions, highlighting geographical specificity as a limitation. Future studies should consider a broader geographical scope to increase the applicability of the results. In addition, the reliance on self-reported data may introduce social desirability bias.

It is important to recognise that the study had some limitations. The results should be interpreted with caution. The baseline level of health literacy was found to be low, and the study design (cross-sectional) does not allow to establish causal relationships, as only some of the potential determinants/factors associated with the level of health literacy among higher education students in Alentejo were analysed. Nevertheless, it is possible to identify a potential health vulnerability. Previous research has shown that health literacy interventions in higher education can significantly improve students' ability to make informed health-related decisions, thereby reducing their vulnerability to poor health outcomes. [50]. The implementation of training programmes focused on healthcare, disease prevention and health promotion could significantly improve the health literacy of higher education students in Alentejo.

Despite these limitations, the study provides valuable insights into the health literacy of higher education students in Alentejo. By addressing these biases and limitations in future research, we can improve the accuracy and applicability of the findings, ultimately leading to more effective health literacy interventions.

**Future research and interventions.** The findings highlight the urgent need for health literacy interventions in higher education, particularly for students at risk of making poor health decisions. Educational programmes focused on healthcare, disease prevention and health promotion could significantly improve health literacy.

Future studies should incorporate qualitative methods to gain deeper insights into the factors that influence health literacy. Policy makers should use these findings to develop targeted strategies to address gaps, particularly among underrepresented groups. Although this study provides valuable insights, it must be interpreted with caution due to potential biases and limitations. Addressing these issues in future research will improve understanding and lead to more effective health literacy strategies.

As expected, health students, older students, those living with health professionals and those with enough money for expenses showed higher levels of health literacy.

## Conclusions

The study identified lower levels of health literacy among higher education students in the Alentejo during the 2022/2023 academic year, highlighting significant challenges and thus answering the first research question. Key determinants of health literacy were identified, including age, presence/absence of chronic disease, course type, living with health professionals, availability of money for expenses and perceived health status. Younger students, those with chronic conditions, those not enrolled in health-related courses, those not living with a health professional, those with limited financial resources and those who perceived their health as unsatisfactory had lower levels of health literacy. These findings answered the second research question. The deficits identified may affect their wellbeing and future health outcomes. Understanding these determinants is crucial for the development of targeted interventions and educational programmes to improve health literacy and promote better health outcomes among higher education students in the region.

Given their significant social influence, higher education students, play a vital role in shaping health-related decisions for themselves and their communities, particularly in a region facing a growing ageing population that poses challenges to health care and social cohesion. The results show significant differences in health literacy, indicating difficulties in understanding health information and making informed decisions. Comparison of these findings with data from other countries can further inform strategies to improve the health and well-being of higher education students. Improving socio-economic support, reducing inequalities and preventing student attrition are essential steps in addressing these challenges.

Overall, the study indicates the need for higher education institutions to prioritise the improvement of health literacy. By identifying specific factors influencing health literacy levels in this population, the study enriches the existing literature and provides important information for educators, policy makers and health professionals involved in decision-making processes within higher education. Efforts should focus on promoting all domains of health literacy: health care, disease prevention and health promotion. This study emphasises the need to integrate health literacy into educational curricula and policy frameworks to effectively address these gaps.

Based on the results of the study, targeted health literacy interventions should be implemented to meet the identified needs of higher education students in the Alentejo region. For example, the implementation of interactive workshops on navigating health systems and promoting healthy lifestyles could improve students' understanding and decision-making about their health. In addition, policy changes should prioritise the integration of health literacy into higher education curricula to ensure comprehensive health education.

The limitations of the study highlight the need for caution in interpreting the results. Reliance on self-reported data and potential bias in participant selection may have influenced the results. Future research should use longitudinal approaches to track changes in health literacy over time and assess the effectiveness of different intervention strategies. These findings are essential to refine health literacy interventions and ensure that they meet the diverse needs of higher education students.

In conclusion, the results of this study highlight the urgent need to address the low levels of health literacy among higher education students in the Alentejo region. To mitigate the risks associated with low health literacy, it is essential to implement targeted interventions that foster informed decision-making and promote healthier lifestyles. Educational institutions should consider developing comprehensive curricula that include health literacy as a core competency, as well as offering practical workshops and establishing support networks involving health professionals.

In addition, future research should aim to use mixed methods to gain deeper insights into the dynamics of health literacy, expand geographically to improve comparability, and use longitudinal approaches to track changes over time. By prioritising health literacy in education and public health policy, we can empower the students with the skills necessary to improve health outcomes and ultimately contribute to a healthier society.

## Author Contributions

**Writing – original draft:** Jorge Rosário, Sara Simões Dias, Sónia Dias, Ana Rita Pedro.

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
