## [Decision Letter · Decision Letter 0]

25 Jun 2024

PONE-D-24-22935Health literacy and its determinants among higher education students in the Alentejo region of southern Portugal - a cross-sectional surveyPLOS ONE

Dear Dr. Rosário,

Thank you for submitting your manuscript to PLOS ONE. After careful consideration, we feel that it has merit but does not fully meet PLOS ONE’s publication criteria as it currently stands. Therefore, we invite you to submit a revised version of the manuscript that addresses the points raised during the review process.

We look forward to receiving your revised manuscript.

Kind regards,

André Ramalho, PhD

Academic Editor

PLOS ONE

Journal Requirements:

2. We note that your Data Availability Statement is currently as follows: 

"All relevant data are within the manuscript and its Supporting Information files."

Reviewers' comments:

Reviewer's Responses to Questions

**Comments to the Author**

1. Is the manuscript technically sound, and do the data support the conclusions?

Reviewer #1: Yes

Reviewer #2: Partly

2. Has the statistical analysis been performed appropriately and rigorously? 

Reviewer #1: Yes

Reviewer #2: No

3. Have the authors made all data underlying the findings in their manuscript fully available?

Reviewer #1: Yes

Reviewer #2: Yes

4. Is the manuscript presented in an intelligible fashion and written in standard English?

Reviewer #1: Yes

Reviewer #2: No

5. Review Comments to the Author

Reviewer #1: The text has been described the relevants contexts. The aims is conected with methodology and results. The discussion is conected with results and presents a complete text. Congratulations for the authors.

Reviewer #2: Detailed Section-by-Section Review of the Manuscript

Manuscript Title:

Health literacy and its determinants among higher education students in the Alentejo region of southern Portugal - a cross-sectional survey

1. Introduction

• Lack of Contextual Background: The introduction provides general information on health literacy but lacks a detailed context specific to the Alentejo region or higher education students in Portugal.

• Limited Discussion on Previous Research: While there is a mention of health literacy's importance, there is sparse discussion of previous studies directly comparing or contrasting this study's findings with existing data.

• Theoretical Framework Missing: The introduction lacks a theoretical framework or model that guides the investigation of health literacy determinants.

• Insufficient Justification for Study: The rationale for why this study is needed in the Alentejo region is not adequately justified.

• Lack of Hypotheses: No hypotheses are presented, which makes it challenging to understand the expected outcomes or the direction of the research.

2. Methodology

• Unclear Sampling Method: The sampling method is not described in detail, making it difficult to assess the representativeness of the sample. Provide a more detailed explanation of the sampling process, including any stratification or weighting applied to ensure representativeness.

• Lack of Detail on Data Collection: The data collection process is not fully detailed, including how the survey was administered and how responses were recorded.

Discuss potential biases introduced by the online collection method and steps taken to mitigate these issues.

3. Results

Problems:

• Overemphasis on Descriptive Statistics: The results section heavily focuses on descriptive statistics without sufficient emphasis on inferential statistics that address the research questions. More detailed analysis, including regression models and other inferential statistics, is needed to identify determinants of health literacy.

• Lack of Contextual Interpretation: Results are presented without sufficient interpretation in the context of the study's aims and existing literature.

4. Discussion

• Overgeneralization: The discussion tends to generalize the findings without acknowledging the specific context of the Alentejo region or higher education students.

• Lack of Critical Evaluation: There is insufficient critical evaluation of the study's findings, including potential biases, limitations, and implications for future research.

5. Conclusion

• Overly General Conclusions: The conclusions drawn are too general and do not adequately reflect the specific findings of the study.

o Example: "This study will contribute to the existing literature by elucidating the health literacy level among higher education students in Alentejo, Portugal, and its determinants."

• Lack of Specific Recommendations: The conclusion does not provide specific recommendations for interventions or policy changes based on the findings.

Recommendations:

• Provide a concise summary of the key findings and their implications.

• Offer specific recommendations for health literacy interventions and policy changes based on the findings.

• Summarize the study's limitations and discuss their implications for future research and practice.

6. PLOS authors have the option to publish the peer review history of their article (what does this mean?). If published, this will include your full peer review and any attached files.

Reviewer #1: No

Reviewer #2: No

---

## [Author Response · Author response to Decision Letter 0]

23 Jul 2024

Response to reviewer 1:

Dear reviewer,

Manuscript title: Health literacy and its determinants among higher education students in the Alentejo region of southern Portugal - a cross-sectional survey

Thank you very much for your feedback on our manuscript. We appreciate your acknowledgement of our work, entitled “Health literacy and its determinants among higher education students in the Alentejo region of southern Portugal: a cross-sectional study”. Below we outline the main improvements we have made to further improve the manuscript:

1. Introduction

• Contextual background: We extended the contextual background to include the specific challenges and characteristics of the Alentejo region and the higher education context in Portugal.

• Literature Review: The discussion of previous studies on health literacy was expanded to provide a more comprehensive overview.

• Theoretical framework: A theoretical framework based on Sørensen et al. (2012) was introduced to better guide the investigation of the determinants of health literacy.

• Rationale: The rationale for the study was strengthened by highlighting gaps in the current literature and the specific challenges faced in the Alentejo region.

• Research questions: We have added specific research questions to clarify the expected outcomes and direction of our research.

2. Methodology

• Sampling procedure: A more detailed explanation of the sampling process was provided, including the selection criteria for higher education institutions in the Alentejo region.

• Data Collection: We have expanded the description of the data collection process, detailing the administration of the online survey and the measures taken to mitigate potential biases.

• Sample robustness: We clarified the robustness of the sample, including the margin of error, which is 3%.

3. Results section

• Inferential statistics: The results section has been expanded to include more detailed inferential statistics, to address the research questions and identify determinants of health literacy more comprehensively.

• Interpretation: We provided a more robust interpretation of the results in relation to the study aims and existing literature.

4. Discussion

• Contextual analysis: The discussion was broadened to avoid over-generalisation, focusing specifically on the context of the Alentejo region and higher education students.

• Comparative analysis: We have enriched the discussion by making explicit comparisons with other relevant studies on health literacy.

• Implications: The discussion of practical and policy implications was extended, with a focus on the Alentejo region.

• Limitations and future research: We included a critical evaluation of the study's limitations and potential biases, as well as suggestions for future research directions.

5. Conclusion

• Summary and recommendations: The conclusion has been revised to provide a clearer summary of the main findings and their implications for health literacy promotion and policy development in the Alentejo region. We have included specific recommendations for interventions and policy changes based on the study findings to ensure actionable and relevant conclusions for stakeholders.

We hope that these improvements will further enhance the clarity, context and impact of our manuscript. Thank you once again for your encouraging feedback.

Sincerely,

Jorge Rosário

Sara Simões Dias

Sónia Dias

Ana Rita Pedro

Response to reviewer 2:

Dear reviewer,

Manuscript Title: Health literacy and its determinants among higher education students in the Alentejo region of southern Portugal - a cross-sectional survey

Thank you for your detailed and constructive feedback on our manuscript. We have made significant revisions based on your comments:

1. Introduction

• Lack of Contextual Background: The introduction provides general information on health literacy but lacks a detailed context specific to the Alentejo region or higher education students in Portugal.

We appreciate your comment about the need for more specific contextual background in relation to the Alentejo region and higher education students in Portugal. In response, we have expanded the introduction to provide a more detailed context that situates our study within the regional and educational contexts of the Alentejo.

The following information has been added:

“Transitioning from a national perspective to a more focused regional context, the Alentejo faces particular challenges. It has the lowest percentage of tertiary educated and high drop-out rates (29). The Alentejo region in southern of Portugal, which is classified both as a Territorial Level 2 (TL2) and a Nomenclature of Territorial Units for Statistics II (NUTS II) region, covers an area of 31,605 km², about 30% of the national territory. This region has a low population density and is threatened by depopulation due to a decreasing young population, an increasing elderly population and low immigration (30,31). Addressing these inequalities requires promoting health literacy and healthy behaviours from an early age, particularly in rural areas facing demographic challenges (32,33). 

The regional focus not only demonstrates the existence of educational and health inequalities but also emphasis the pressing necessity to conduct research and implement solutions at the local level. By elucidating the health literacy levels of higher education students in Alentejo, this research can provide insights into regional inequalities and inform policies aimed at reducing these disparities, ultimately benefiting both individual and community health outcomes.

It is therefore imperative to assess the level of health literacy among higher education students and investigate its association with various determinants. To the best of our knowledge, the current level of health literacy among higher education students in Alentejo is unknown, which makes it challenging to design effective health interventions.”

• Limited Discussion on Previous Research: While there is a mention of health literacy's importance, there is sparse discussion of previous studies directly comparing or contrasting this study's findings with existing data.

We have revised the introduction to include a more comprehensive discussion of previous studies on health literacy, specifically addressing how our findings compare and contrast with existing data in the field.

• Theoretical Framework Missing: The introduction lacks a theoretical framework or model that guides the investigation of health literacy determinants.

A theoretical framework guiding our investigation of the determinants of health literacy has been included in the introduction, providing a clearer basis for our study

The following information has been added:

“In order to gain further insight into the pathways linking health literacy to health behaviours and outcomes, Sørensen et al. (2012) developed the Integrated Model of Health Literacy. This theoretical framework delineates the pathways linking health literacy to health behaviours and outcomes. The model identifies the determinants and factors influencing health literacy levels (the antecedents) and their subsequent health outcomes (the consequences) (1). This theoretical framework, predominant in Europe, describes four core competencies of health literacy: access (the ability to seek, find, and obtain health information), understand (the ability to comprehend accessed health information), appraise (the ability to interpret, filter, judge and evaluate the health information that was assessed) and apply (the ability to communicate and utilise the information to make decision to maintain and improve health). The four competencies, namely accessing, understanding, appraising and applying, can be applied across the three domains of the health continuum, including healthcare, disease prevention and health promotion (1). 

An understanding of these competencies, serve to highlight the dynamic and holistic nature of health literacy. This encompasses not only the abilities of individuals, but also the inherent complexity of health systems. This framework facilitates comprehension and improvement of health literacy across heterogeneous populations and healthcare settings. The empowerment of individuals through the acquisition of these skills fosters autonomy and informed decision-making in matters pertaining to health (6) thereby enhancing social participation and well-being. Furthermore, the structural organisation and availability of resources have a significant impact on health literacy levels and individuals' ability to make informed health decisions, which in turn affect personal and community health (7).”

• Insufficient Justification for Study: The rationale for why this study is needed in the Alentejo region is not adequately justified.

We have strengthened the rationale for our study in the Alentejo region by highlighting specific gaps in the current literature and emphasising the unique regional challenges that justify our research focus.

The following information has been added: 

“It is therefore imperative to assess the level of health literacy among higher education students and investigate its association with various determinants. To the best of our knowledge, the current level of health literacy among higher education students in Alentejo is unknown, which makes it challenging to design effective health interventions. “

• Lack of Hypotheses: No hypotheses are presented, which makes it challenging to understand the expected outcomes or the direction of the research.

We included the following research questions: 

• “What is the level of health literacy among higher education students in the Alentejo region of southern Portugal?

• What is the relationship between health literacy and determinants such as socio-demographic variables, the presence of chronic disease, perceived health status, and perceived availability of money for expenses among higher education students in the Alentejo region?

The responses to these questions will facilitate the implementation of targeted interventions and policies designed to reduce inequalities and enhance health outcomes in this region. It is expected that health students, younger students who live with health professionals and have enough money for expenses will have higher health literacy.”

2. Methodology

• Unclear Sampling Method: The sampling method is not described in detail, making it difficult to assess the representativeness of the sample. Provide a more detailed explanation of the sampling process, including any stratification or weighting applied to ensure representativeness.

For a better understanding, we have explained our sampling method. Our sample was selected from higher education institutions in the Alentejo region, specifically in Portalegre, Santarém, Évora and Beja. We identified our target population as students enrolled in higher education institutions in the Alentejo region.

We didn't use any stratification or weighting in our study. However, our sampling method was designed to capture a broad and diverse subset of students. Our aim remained to gain an overall understanding of the student population in the Alentejo region.

No stratification or weighting was applied in our sampling process, as the objectives of our study are specifically centred on this region and its higher education institutions. Our aim is not to generalise the results beyond this context. We don't intend to generalise the results to every student at every institution. The sample has a margin of error of 3%, which is low in most cases and guarantees the robustness of our results.

• Lack of Detail on Data Collection: The data collection process is not fully detailed, including how the survey was administered and how responses were recorded.

Discuss potential biases introduced by the online collection method and steps taken to mitigate these issues.

We have expanded the description of our data collection process, including how the survey was administered online and the steps taken to mitigate potential bias introduced by this method.

The following information has been added:

“Institutional Approval and Promotion

The governing body of each institution was contacted via email to explain the aims and importance of the study, and to request their approval. Once approval was granted, the institution promoted the link to the questionnaire, along with the consent form, to students via their official social media platforms and email lists. To reinforce the importance of participation, and encourage participation, course coordinators were asked to highlight the importance of the survey in emails and meetings. This multi-channel approach was intended to ensure high visibility and maximise response rates.

Survey Administration

Responses were collected electronically using an online survey tool that automatically recorded and securely stored data. The tool ensured anonymity and confidentiality in line with ethical research standards. Regular reminders were sent to students to maximise participation, and technical support was provided to resolve any problems encountered when completing the survey. The survey was designed to be mobile-friendly, ensuring accessibility on various devices, including smartphone or tablets. It was conciseness to minimise the time required for completions.

Addressing Potential Biases

The potential biases inherent in the online data collection method were recognised, including the exclusion of students without internet access, electronic devices or familiarity with digital technologies. To mitigate these concerns, the survey was made accessible on various devices available at higher education institutions, and participants were provided with technical support available at institutions where necessary. In addition, the multi-channel promotion strategy aimed to widen participation across different student populations, thus ensuring a more diverse and representative sample. “

3. Results

Problems:

• Overemphasis on Descriptive Statistics: The results section heavily focuses on descriptive statistics without sufficient emphasis on inferential statistics that address the research questions. More detailed analysis, including regression models and other inferential statistics, is needed to identify determinants of health literacy.

In response to your feedback, we have enhanced the results section with more detailed inferential statistics, such as regression models, to better address the research questions and identify determinants of health literacy.

It was added the following information:

“The multiple linear regression model presented in Table 4 shows the relationship between health literacy and several independent variables among higher education students in the Alentejo region. The independent variables included were age, presence of chronic disease, attendance in health-related courses, living with a health professional, availability of money for expenses, perceived health status, and academic institution. The intercept represents the baseline health literacy score.

The results show that younger students, particularly those aged 16-20 and 21-25, had significantly lower health literacy scores than those aged 26-30, with differences of 3.5 and 3.4 points respectively (p < 0.001). The absence of chronic disease was associated with a higher health literacy score of 1.8 points (p = 0.002), suggesting better health literacy among students without chronic disease. Not attending a health-related course was associated with a significantly lower health literacy score of 9.8 points (p < 0.001), highlighting the impact of health education on health literacy levels.

The data also show that not living with a health professional was associated with a lower health literacy score of 4.2 points (p < 0.001), suggesting that living with health professionals has a positive impact on health literacy. Poor availability of money for expenses was associated with a lower health literacy score by 6.7 points (p < 0.001), highlighting the role of economic factors in health literacy. In addition, perceiving one's health status as unsatisfactory was associated with a lower health literacy score by 4.3 points (p < 0.001), indicating a relationship between sel

---

## [Decision Letter · Decision Letter 1]

20 Aug 2024

Health literacy and its determinants among higher education students in the Alentejo region of southern Portugal - a cross-sectional survey

PONE-D-24-22935R1

Dear Dr. Rosário,

We’re pleased to inform you that your manuscript has been judged scientifically suitable for publication and will be formally accepted for publication once it meets all outstanding technical requirements.

Kind regards,

André Ramalho, PhD

Academic Editor

PLOS ONE

Reviewers' comments:

Reviewer's Responses to Questions

**Comments to the Author**

1. If the authors have adequately addressed your comments raised in a previous round of review and you feel that this manuscript is now acceptable for publication, you may indicate that here to bypass the “Comments to the Author” section, enter your conflict of interest statement in the “Confidential to Editor” section, and submit your "Accept" recommendation.

Reviewer #3: All comments have been addressed

2. Is the manuscript technically sound, and do the data support the conclusions?

Reviewer #3: Yes

3. Has the statistical analysis been performed appropriately and rigorously? 

Reviewer #3: Yes

4. Have the authors made all data underlying the findings in their manuscript fully available?

Reviewer #3: Yes

5. Is the manuscript presented in an intelligible fashion and written in standard English?

Reviewer #3: Yes

6. Review Comments to the Author

Reviewer #3: (No Response)

7. PLOS authors have the option to publish the peer review history of their article (what does this mean?). If published, this will include your full peer review and any attached files.

Reviewer #3: **Yes: **Bruno Filipe Coelho da Costa

---

## [Editor Report · Acceptance letter]

13 Sep 2024

PONE-D-24-22935R1 

PLOS ONE

Dear Dr. Rosário, 

I'm pleased to inform you that your manuscript has been deemed suitable for publication in PLOS ONE. Congratulations! Your manuscript is now being handed over to our production team.

Kind regards, 

on behalf of

Prof. Dr. André Ramalho 

Academic Editor

PLOS ONE